# Genome-Wide Identification and Expression Analyses of the MADS-Box Gene During Flowering in *Primulina huaijiensis*

**DOI:** 10.3390/plants14121843

**Published:** 2025-06-16

**Authors:** Jie Zhang, Xinxia Cai, Qin Liu, Ziyi Lei, Chen Feng

**Affiliations:** 1Jiangxi Provincial Key Laboratory of Ex Situ Plant Conservation and Utilization, Lushan Botanical Garden, Chinese Academy of Sciences, Jiujiang 332900, China; zhangjie@lsbg.cn (J.Z.); caixinxia2983@163.com (X.C.); liuqin232022@163.com (Q.L.); leiziyi111@outlook.com (Z.L.); 2School of Life Sciences, Nanchang University, Nanchang 330031, China

**Keywords:** *Primulina huaijiensis*, MADS-box, gene family, phylogenetic analysis, expression patterns

## Abstract

*Primulina huaijiensis* is a promising candidate for eco-bottle flowers, yet the genes related to flowering remain unexplored despite the availability of genomic data for several years. *MADS-box* genes constitute a large family of transcription factors that play crucial roles in plant growth and development, particularly in flower development. In this study, we identified 84 *MADS-box* genes (*PhuMADS*) in *P. huaijiensis* genome and analyzed their evolution and expression profiles to gain insights into the flowering mechanism. The 84 genes constitute 29 type I and 55 type II *MADS-box* genes. Phylogenetic analysis further classified them into 17 subfamilies, which were randomly distributed across 18 chromosomes and four scaffolds. *PhuMADS* genes exhibit a range of 1 to 12 exons and share conserved motifs. Segmental duplication was found to be the primary driver of *PhuMADS* gene family expansion, with duplicated gene pairs undergoing purifying selection. *Cis*-acting elements analysis revealed *PhuMADS* promoters harbor abiotic stress-, hormone-, light-, and growth-related motifs, implicating roles in development and environmental adaptation in *P. huaijiensis*. RNA-seq showed distinct expression patterns of *PhuMADS* genes among different tissues or developmental stages. The results of qRT-PCR analysis of selected genes further validated the RNA-seq findings, suggesting these genes may exert distinct functional roles during floral development. This study laid a theoretical foundation for further functional studies of the *MADS-box* genes in *P. huaijiensis*.

## 1. Introduction

MADS-box transcription factors are widely present in plants, animals, and fungi, and play important roles in plant growth and development cycle, including regulation of plant abiotic stress response, flowering time regulation, floral organ identity, seed development, and fruit ripening [1,2,3]. *MADS-box* is an abbreviation of four genes initials: *Minichromosome maintenance 1* (*MCM1*) of *Saccharomyces cerevisiae* [4], *AGAMOUS* (*AG*) of *Arabidopsis thaliana* [5], *DEFICIENS* (*DEF*) of *Antirrhinum majus* [6], and the *Serum response factor* (*SRF*) of *Homo sapiens* [7]. Based on gene structure and molecular phylogenetic analysis *MADS* gene can be divided into two major categories, which are defined as type I and type II [8]. The MADS-domain of type I protein in plants can be further subdivided into three groups: Mα, Mβ, and Mγ. The type II genes, also known as MIKC (MADS Intervening Keratin-like C-terminal) genes, are further classified into MIKC^C^ and MIKC* subfamily [9,10].

Previous studies have revealed that MADS-box proteins contain four domains: the MADS-box domain (M), the intervening domain (I), the Keratin-like domain (K), and the C-terminal domain (C) [11]. Type II proteins contain all four of these domains, while type I proteins lack the K domain [12]. Among them, the MADS-box domain is the most highly conserved and present in all *MADS-box* genes, while the K domain is the second most conserved domain that characterized by a coiled–coil structure [13,14]. In contrast, the C domain is the most variable and is involved in protein complex formation and transcriptional activation [15]. The intervening I domain that contributes to DNA binding specificity and dimerization is relatively conserved compared to the C domain [16,17].

The *MADS-box* genes play an important role in plant growth and development, especially in flower development. They are crucial developmental regulators of sepals, petals, stamens, and carpels [18]. The classical ABC model proposed that the development of the floral organ was controlled by certain genes [19]. This model was subsequently expanded to the ABCDE model, providing a more comprehensive framework for understanding floral organogenesis. According to this model, A-class and E-class genes are involved in sepal development, while petal development is jointly regulated by A, B, and E-class genes. The formation of stamens is co-regulated by B, C, and E-class genes, whereas the carpels are jointly controlled by C and E-class genes. Ovule development is governed by C, D, and E-class genes [20]. Additionally, the *MADS-box* genes also play a key role in abiotic stress response, reproductive development, and so on. For instance, overexpress *OsMADS25* enhances salt stress tolerance in rice and *A. thaliana*, the interaction of *AGL11* and *MBP3* influence locule gel formation and lead the all-flesh type tomato [21,22].

*Primulina huaijiensis* Z.L. Ning and J. Wang, an evergreen perennial herb in the Gesneriaceae family, is endemic to Huaiji county in northwest Guangdong province, China, where it typically grows on wet rocks in limestone caves [23]. It is a micro-endemic species with the smallest genome size in the *Primulina* genus [24]. It features small white flowers and is a promising candidate for eco-bottle flowers for a dwarf plant. These characteristics make it a model species for studying the survival strategies of endangered plants in species habitats. While extensive studies have been conducted on the *MADS-box* genes in plants, studies on *P. huaijiensis* remain limited. The sequenced *P. huaijiensis* genome provides an opportunity to analyze the detailed *MADS-box* genes. In this study, we identified and analyzed the *MADS-box* genes and identified several genes that may be associated with flowering through transcriptomic and RT-PCR approaches. Our findings contribute to a better understanding of the *MADS-box* genes in *P. huaijiensis* and provide valuable insights into their function in *P. huaijiensis*.

## 2. Results

### 2.1. Identification and Physicochemical Property Analysis of MADS-Box Gene Family in P. huaijiensis

According to the results of the BLAST alignment and the hidden Markov model (HMM) of the SRF-TF domain (PF00319), MEF2_binding (PF09047) and the K-box domain (PF01486), 101 *MADS-box* candidate genes were screened from the whole genome of *P. huaijiensis*. Based on the *MADS-box* genes of *A. thaliana*, a total of 84 domain-intact MADS-box proteins were identified in *P. huaijiensis* and designated as PhuMADS1 to PhuMADS84 based on their chromosomal locations (Appendix A). The analysis of the proteins’ physical and chemical properties showed that 54 proteins ranged in length from 200 to 300 amino acids, 20 proteins consisted of 100 to 200 amino acids, 7 proteins exceeded 300 amino acids, and only 3 proteins were shorter than 100 amino acids (Appendix A). PhuMADS38 was identified as the shortest protein with only 62 amino acids, while PhuMADS84 was the longest one with 396 amino acids. The molecular weights of the PhuMADS proteins ranged from 7054.3 (PhuMADS38) to 45677.78 (PhuMADS84) Da. The theoretical isoelectric point (pI) ranged between 4.79 (PhuMADS64) and 10.22 (PhuMADS10). Among the MADS-box proteins, 64 were classified as alkaline with pI values greater than 7.5, while 14 proteins were acidic with pI values less than 6.5, and only 6 proteins had a PI value between 6.5 and 7.5. The instability indexes for most of the PhuMADS proteins were greater than 40, except for PhuMADS5, PhuMADS7, PhuMADS22, PhuMADS23, PhuMADS56, and PhuMADS57. The aliphatic index analysis indicated that most of the MADS-box proteins were hydrophilic with an aliphatic index of less than 100, with the exception of PhuMADS78. The grand average of the hydropathicity of all MADS-box proteins ranged from −0.918 (*PhuMADS73*) to −0.279 (*PhuMADS78*), further confirming their hydrophilic features. Subcellular localization predictions suggested that all PhuMADS proteins were located in the nucleus.

### 2.2. Classification and Phylogenetic Analysis of the PhuMADS

A Maximum Likelihood (ML) tree was constructed using the full-length protein sequences of the identified 84 *PhuMADS* genes to clarify the evolutionary relationship of *MADS-box* in *P. huaijiensis*. Based on the grouping of MADS-box proteins in *A. thaliana*, the *MADS-box* proteins of *P. huaijiensis* were classified into two types and 17 subfamilies (Figure 1). The 84 *PhuMADS* genes included 29 type I genes and 55 type II genes. The type I *PhuMADS* genes were further divided into three subfamilies: Mα (18 genes), Mβ (3 genes), and Mγ (8 genes). The type II genes were classified into two clades: the MIKC* (4 genes) and MIKC^c^ (51 genes) clades. The MIKC* clade included two subfamilies: MICK*-P (two genes) and MICK*-S (two genes). In contrast, the MIKC^c^ included 12 subfamilies: SVP (six genes), AGL15/18 (AGAMOUS-like 15/18, three genes), AGL17/ANR1 (AGAMOUS-like 17/ANAEROBIC RESPONSE 1, two genes), Bsister (two genes), AP3/PI (APETALA 3/PISTILATA, six genes), FLC (FLOWERING LOCUS C, one gene), SOC1 (SUPPERSSOR OF OVEREXPRESSION OF CO 1, six genes), AP1/FUL (APETALA 1/FRUITFULL, seven genes), SEP (SEPALLATA, five genes), AGL6 (AGAMOUS-like 6, six genes), AGL12 (AGAMOUS-like 12, one gene), and AG/STK (AGAMOUS/SEEDSTICK, six genes). Notably, AGL6, AP1/FUL, SVP, and AP3/PI subfamilies are significantly expanded in *P. huaijiensis* compared with *A. thaliana*, while FLC, Mα, Mβ, and Mγ subfamilies are significantly contracted. Previous studies showed that *AP1*, *AGL6*, and *SEP* were involved in floral organ identity and floral termination functions in *Petunia* × *hybrida* and *Epimedium sagittatum* [25,26], while *FLC*, *SVP*, and *SOC1* were the core regulators in flowering pathway [27]. Notably, the genes of *AGL6*, *AP1*/*FUL*, *SEP*, *AP3*/*PI*, *FLC*, *SVP*, and *SOC1* have two, four, six, two, six, two, and six members in *A. thaliana*, respectively, while they have six, seven, five, six, one, six, and six members in *P. huaijiensis*, respectively [28]. The number of those genes in *P. huaijiensis* was greater than *A. thaliana*. This may suggest that the flower morphogenesis and flowering regulatory network in *P. huaijiensis* was more complex and diverse than *A. thaliana*. Additionally, the FLC subfamily has been identified as a key component to respond to vernalization in *A. thaliana* [29]. The FLC subfamily is missing in many plants, such as cucumber [30], chayote [31], barley [32], and rice [33]. This may indicate that *P. huaijiensis* probably require vernalization.

### 2.3. Chromosomal Localization of the PhuMADS

The 84 *PhuMADS* genes were unevenly distributed on 18 chromosomes (Figure 2A). The number of *PhuMADS* genes on each chromosome ranged from one to ten (Figure 2B). In most chromosomes, the number of type I genes is lower than that of type II genes, with a few of exception on chromosome 11, 14, 17, and 18. The proportion of genes on each chromosome ranged from 1% to 12% (Figure 2C). Chr11 had the maximum number of *PhuMADS* genes (10), accounting for 12%, whereas Chr16 had only one *MADS-box* gene, accounting for 1% (Figure 2B,C). No chromosomal bias was observed in the distribution of type II genes with an exception on Chr16 that lacked type II genes (Figure 2B, Appendix A).

### 2.4. Conserved Motif and Gene Structure Analysis of PhuMADS

We identified 20 conserved motifs, which labeled motifs 1 to 20 (Figure 3B and Appendix A). PhuMADS proteins in the same group had similar motifs. Motif 1 contains a conserved 41 amino acids domain and present in all PhuMADS proteins at the N-terminus. Motif 3, consisting of 11 amino acids, is found in 77 of these proteins, also at the N-terminus. Motifs 1 and 3 composed the classic MADS domain, and were widely present in many plants [34]. A total of 77 PhuMADS had motif 3 that was located on the N-terminus comprising 11 amino acids. However, some motifs were found only in certain subfamilies. For example, motifs 9 and 13 were only found in the Mα subfamily, while motif 11 and motif 12 were specifically found in the Mβ subfamily and Mγ subfamily, respectively.

The UTR/CDS arrangements results revealed the diversity among *PhuMADS* genes. Notably, genes in the same subfamily exhibited similar structures (Figure 3C). The number of exons varied from 1 to 12, with *PhuMADS84* having the largest count of 12 exons. Furthermore, type II genes (MIKC^c^ and MIKC*) contained multiple introns, whereas type I genes (Mα, Mβ, and Mγ) usually had either none of introns or a single intron. Overall, the average intron number of Type II genes was significantly much higher than those of type I genes. Additionally, the length of introns varied greatly among genes. Some members in the same phylogenetical clade exhibited distinct intron/exon arrangements. For example, *PhuMADS75* in the AGL15/18 subfamily contains only two exons, whereas *PhuMADS32* and *PhuMADS33* have nine exons.

### 2.5. Collinearity Analyses of the PhuMADS Within and Between Species

Among *PhuMADS* genes, only one gene pair was found as tandem duplication gene pairs (*PhuMADS12* and *PhuMADS8*, Appendix A). Interestingly, the Ka/Ks value of the gene pair was greater than 1, indicated that these genes were undergoing the positive selection. Additionally, a total of 51 segmental duplication gene pairs were also found (Figure 4A, Appendix A), suggesting that segmental duplication was the primary driving force for the expansion of the *PhuMADS* gene family. Notably, most gene pairs were clustered within the same subfamily (Appendix A). Subsequently, we calculated Ka/Ks ratios to investigate the evolutionary pressures on the orthologous of *MADS-box* gene pairs. The results showed that all the segmental duplication gene pairs exhibited a Ka/Ks < 1 (Appendix A), indicating that these *PhuMADS* genes were under purifying selection pressure during evolution.

To further analyze the orthologous relationships of *MADS-box* genes between *P. huaijiensis* and other representative species (including *Arabidopsis thaliana*, *Primulina eburnea*, *Populus trichocarpa*, *Vitis vinifera*, and *Zea mays*, Figure 4B), we conducted a whole genome-wide syntenic analysis. A total of 186 pairs of orthologous genes were identified (Figure 3B, Appendix A). Among them, the highest number of collinear gene pairs were found between *P. eburnea* (73), followed by *P. trichocarpa* (48), *V. vinifera* (30), *A. thaliana* (28), and *Z. mays* (7). Furthermore, *PhuMADS14*, *PhuMADS17*, and *PhuMADS49* have collinear relationships with all these five species, which indicates that these genes may be conserved in function.

### 2.6. Cis-Element Analysis of PhuMADS

To gain insights into the regulatory mechanisms of *PhuMADS* genes expression, *cis*-acting elements were analyzed. A total of 64 *cis*-acting elements have been found, which are divided into six categories: development related elements (9), environmental stress-related elements (6), hormone responsive elements (11), light responsive elements (30), promoter related elements (3), and site-binding related elements (5). Among them, light responsive elements were the most abundant, comprising 48.6% (992) of the total number (Figure 5B, Appendix A). Notably, twenty *PhuMADS* genes (*PhuMADS11*, *PhuMADS74*, *PhuMADS8*, *PhuMADS12*, *PhuMADS44*, *PhuMADS25*, *PhuMADS36*, *PhuMADS38*, *PhuMADS2*, *PhuMADS82*, *PhuMADS56*, *PhuMADS19*, *PhuMADS67*, *PhuMADS75*, *PhuMADS17*, *PhuMADS29*, *PhuMADS41*, *PhuMADS37*, *PhuMADS45*, and *PhuMADS73*) lacked development related elements. All *PhuMADS* genes, except for *PhuMADS36* and *PhuMADS37*, contained a combination of development related elements, environmental stress-related elements, hormone responsive elements, and light responsive elements (Figure 5B).

### 2.7. Expression Profiling of PhuMADS-Box Genes

To further investigate the potential role of the *PhuMADS* genes in flowering, we conducted RNA-seq on flowers of *P. huaijiensis*, including the pistil, stamen, petal, sepal, and flower of five different developmental stages (S1, S2, S3, S4, and S5). The results showed that most *PhuMADS* genes were differentially expressed across different tissues and developmental stages (Figure 6, Appendix A). In general, members in the same subfamily presented similar expression patterns with minor variations. Most members in group AGL6, AGL12, Mγ exhibited high expression levels in the pistil, suggesting that these genes may involve in the pistil formation and flowering. Notably, the majority of *PhuMADS* genes showed a high expression level in S1 or S2 stage, indicating that these genes play a crucial role in flower development and formation in *P. huaijiensis*. For example, the members in AP3/PI subfamily were mainly expressed in stamens and petals, while the members in SEP subfamily were expressed except in stamens. The genes of AP1/FUL subfamily mainly expressed in sepals, and the genes in AG/STK subfamily mainly expressed in pistils.

The transcriptomic analysis revealed distinct expression patterns of these *PhuMADS* genes among different tissues or developmental stages. The majority of type I genes exhibited constitutively high expression levels across pistil and S2 stage, whereas a subset of type II genes showed significantly a high expression in pistil, petal, and S1 to S2 stage (Figure 6). To further characterize and verify specific expression patterns of *PhuMADS* among different stages (S1 to S5 stage), 17 representative genes with high expression level (FPKM > 2) and significant difference between stages were selected to verify their expression level by qRT-PCR (Figure 7). Three distinct expression patterns emerged from this analysis. Firstly, *PhuMADS41*, *PhuMADS6*, and *PhuMADS57* showed a high expression level during early developmental stages (S1 or S2 stage), followed by a progressive down regulation in later stages (Figure 7A–D). This suggests potential involvement floral meristem establishment or differentiation during flowering. Secondly, *PhuMADS9*, *PhuMADS61*, *PhuMADS42*, and *PhuMADS84* showed minimal expression in S1 stage, but sharply highly expression from S2 to S5 stage (Figure 7E–H). Finally, *PhuMADS71*, *PhuMADS21*, *PhuMADS3*, and *PhuMADS63* exhibit low expression level in S1 stage. However, the expression gradually rose along with the development stages, reaching a transient peak before gradually diminishing (Figure 7I–L). Additionally, *PhuMADS16*, *PhuMADS32*, *PhuMADS34*, *PhuMADS1*, and *PhuMADS43* have non-significant differential expression among different developmental stages (Figure 7M–Q). These results are largely consistent with RNA-seq analysis, suggesting these genes may exert distinct functional roles during floral developmental stages.

## 3. Discussion

In recent years, advancements in whole genome sequencing projects have accelerated the study of gene families. Numerous *MADS-box* genes have been identified in various plants, such as *Camellia chekiangoleosa*, *Salvia miltiorrhiza*, and maize [35,36,37]. In this study, we identified 84 *MADS-box* genes in *P. huaijiensis* based on the whole genome sequence. Phylogenetic analysis divided them into two main types and 17 subfamilies. The gene structure analysis showed that the number of introns in type II genes was significantly higher than type I. The distribution of *MADS-box* genes in the genome revealed distinct patterns between type I and type II genes.

FLC is a transcription factor that is the central regulator in the vernalization pathway of the flowering time regulatory network [29]. *P. huaijiensis* has fewer FLC members (only one gene) compared to *Arabidopsis* (six genes). Similarity, research in litchi (*Litchi chinensis*) also showed that it needs low temperature to induce flowering, this may be due to limited FLC members. However, this cold temperature requirement can be reduced or replaced by drought treatment. This suggests that the involvement of alternative regulatory mechanisms may compensate for FLC in the cold-induced flowering pathway [38]. Whether similar compensatory mechanisms exist in *P. huaijiensis* requires further experimental verification.

The K-box domain contained K1, K2, and K3 subdomains, which corresponded to motifs 2, 5, and 7, respectively. All the type II proteins lacking the K-box domains either do not contain any motifs or contained one or two motifs. For instance, the K-box was absent from seven PhuMADS members among the 55 MIKC-type proteins identified in this study. Notably, all members in MIKC* subfamily lacked K-box domains. This feature was also reported in blueberry [39], which possibly related to MIKC* being a class of genes combining both features of type I and type II genes. Additionally, some studies indicated that MIKC^C^ genes may be the most ancient members of *MADS-box* genes, and type I genes probably evolved from MIKC^C^ genes [9]. This suggests that MIKC* genes may represent a transitional class that was retained following the loss of the K-box domain during the evolution of MIKC^C^ genes.

The collinearity analysis identified that 61 *PhuMADS* genes were associated with segmental replication events, while only one gene pair was associated with a tandem replication event. This suggested that both segmental and tandem replication events contributed to the *PhuMADS* expansion in *P. huaijiensis*, with segmental duplication events likely acting as the primary driving force. In contrast to previous studies, which reported higher frequencies of duplication events among type I *MADS-box* genes in species like Arabidopsis and barley, our study found that more type II genes were involved in duplicated segments compared to type I genes.

In general, this study summarizes *PhuMADS* genes associated with flowering and floral development, outlines their fundamental characteristics, and validates several genes potentially linked to flowering processes. However, the functional characterization of these genes remains insufficient to elucidate their specific roles in distinct flowering stages. These genes can serve as key targets for subsequent functional research and be rapidly modified via gene editing to flowering. These genes may also be the key factor for the domestication and improvement of *P. huaijiensis*. This will not only enhance our understanding of gene function in certain flowering processes in plants adapted to special Karst habitats, but also hold significant implications for the conservation of endangered species in the unique Karst habitats.

## 4. Materials and Methods

### 4.1. Plant Materials

The *P. huaijiensis* plants used in this study were sampled from Huaiji country (Huaiji, China) and cultivated at Lushan Botanical Garden, Chinese Academy of Sciences, Nanchang, China (115.8382° E; 28.9112° N). Plants were watered as needed. Flowers at five different developmental stages (S1 to S5) were harvested according to the flower length: S1 (0.1 cm), S2 (0.3 cm), S3 (0.5 cm), S4 (0.7 cm), and S5 (0.9 cm) (Figure 8). Different tissues of the flowers (pistil, stamen, sepal, and petal) during blooming stage were also sampled. Entire flowers from each developmental stage were mixed and harvested. The floral organs, including pistils, stamens, petals, and sepals were also collected from the flowers of stage S1 to S5. These samples were collected and frozen in liquid nitrogen immediately and stored at −80 °C. Each sample included three biological replicates.

### 4.2. Identification of PhuMADS Genes

The genomic and protein data of *P. huaijiensis* were obtained from our previously published data [40]. The MADS-box protein sequences of *A. thaliana* were obtained from The Arabidopsis Information Resource (TAIR) database (https://www.arabidopsis.org/). To identify all candidate *PhuMADS* genes, the MADS-box protein sequences of *A. thaliana* were used as queries to BLAST (v2.15.0) against the *P. huaijiensis*. We also downloaded MADS protein SRF-TF domain (PF00319), MEF2_binding domain (PF09047), and K-box domain (PF01486) from the Pfam website (https://pfam.xfam.org/) to construct a hidden Markov model search (HMM). Then all the candidate protein sequences were assessed based on the presence of the conserved domain via the Conserved Domain Database (CDD) search (https://www.ncbi.nlm.nih.gov/Structure/bwrpsb/bwrpsb.cgi, accessed on 25 July 2024). Sequences that either incorrectly occupied or did not carry an entire domain were removed. In addition, physicochemical properties of PhuMADS proteins, including the number of amino acids, molecular weight, theoretical isoelectric point, instability index, aliphatic index, and grand average of hydropathicity, were evaluated using the TBtools [41]. Additionally, the subcellular localization of the MADS proteins was predicated in the Plant-mPLoc website (http://www.csbio.sjtu.edu.cn/bioinf/plant-multi/, accessed on 22 May 2025).

### 4.3. Phylogenetic Analysis of PhuMADS Genes

To understand the phylogenetic relationship and the classification of the *MADS-box* genes, full-length MADS-box protein sequences of *A. thaliana* and *P. huaijiensis* were aligned using MAFFT in PhyloSuite v1.2.3 software [42,43]. The individual unrooted maximum likelihood (ML) tree of PhuMADS was constructed by IQ-TREE v1.6.12 software with parameters of 1000 bootstraps and automatic model selection [44]. The results were then visualized in FigTree v1.4.4 (http://tree.bio.ed.ac.uk/software/figtree, accessed on 6 December 2024).

### 4.4. Chromosomal Location of PhuMADS Genes

The distribution of *PhuMADS* genes and gene density were extracted and visualized from the General Feature Format (GFF) file in TBtools.

### 4.5. Conserved Motif and Structure Analysis of PhuMADS Genes

The identified PhuMADS protein sequences were submitted to the MEME online project (https://meme-suite.org/meme/, accessed on 26 July 2024) to analyze the conserved motifs [45] within the following paraments: maximum number of motifs was set to 20, and other parameters were set by default. The UTR (Untranslated Regions) and CDS (Coding Sequence) of *PhuMADS* were extracted and visualized with the conserved motif results by TBtools.

### 4.6. Collinearity Analyses of the PhuMADS Within and Between Species

Gene duplication and the collinearity of *PhuMADS* genes across different species were analyzed and visualized using Circos v0.69 and JCVI [46,47]. The protein sequences and GFF files of these species were downloaded from phytozome database (phytozome-next.jgi.doe.gov), including *A. thaliana* (TAIR 11), *Populus trichocarpa* (v4.1), *Zea mays* (B73), and *Vitis vinifera* (v2.1). Synonymous (Ka) and nonsynonymous (Ks) substitutions, as well as their ratios, were also analyzed based on the collinearity result of synteny analysis.

### 4.7. Cis-Element Analysis of PhuMADS

The 2000 bp upstream sequences of 84 *PhuMADS* genes were extracted as the promoter sequence. These sequences were then submitted to the PlantCare online website (https://bioinformatics.psb.ugent.be/webtools/plantcare/html/, accessed on 21 September 2024) to query the *cis*-acting elements of each gene [48]. The results were visualized using TBtools and R v4.3.1.

### 4.8. Expression Profiling of PhuMADS

To investigate the expression patterns of *PhuMADS* genes in different tissues across developmental stages, we performed transcriptome sequencing. The reads were mapped to the reference genome of *P. huaijiensis* using Hisat2 [49], and the transcripts were assembled quantified using the STRINGTIE v2.1.5 [50]. Fragments per kilobase of exon per million fragments mapped (FPKM) were used to estimate the gene expression levels. The relative expression for each gene member across the four tissues were normalized by *z*-score method and visualized in R v4.3.1.

### 4.9. Expression Analysis of PhuMADS in Different Stage

For qRT-PCR analysis, total RNA was extracted from samples using an *EASY*spinPlus Complex Plant RNA kit (Aidlab Biotech, Beijing, China) according to the manufacturer’s instructions. First-strand cDNA was synthesized from 1 µg RNA with One-Step gDNA Removal and cDNA Synthesis SuperMix (TransGen, Beijing, China). qRTC-PCR was performed with PerfectStart Green qPCR SuperMix (TransGen, Beijing, China) on a CFX Connect Real-Time System (Bio-Rad, Hercules, CA, USA). The *P. eburnea GAPDH* gene was used as the internal reference and the relative expression levels of each *MADS* gene were calculated using the 2^−∆∆Ct^ method [51,52]. All qRT-PCRs were performed with three biological and three technical replicates. All statistical analyses in this section were performed with ANOVA (one-way analysis of variance) in R v4.3.1. All primer sequences were designed by Primer Premier 5.0 and listed in Appendix A.

## 5. Conclusions

In this study, we identified 84 *MADS-box* genes from the genome of *P. huaijiensis*, including 29 type I and 55 type II MADS genes. A phylogenetic analysis showed that 84 *PhuMADS* genes were divided into 17 subfamilies. Notably, differences in gene structure, conserved motifs, and *cis*-acting elements among different subfamilies were analyzed among these subfamilies suggesting that type II genes exhibit more complexity and importance than type I genes in *P. huaijiensis*. This research contributes valuable insights into the *MADS-box* gene family and establishes a foundation for further exploration of its evolution in plants.

## Figures and Tables

**Figure 1 plants-14-01843-f001:**
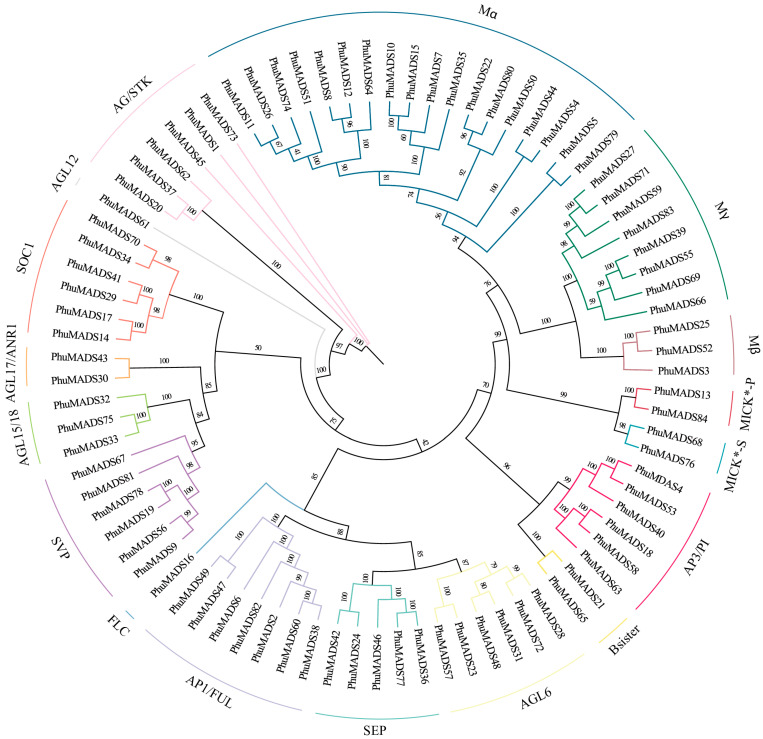
Phylogenetic tree of the PhuMADS. Different colors are used to distinguish 17 subfamilies.

**Figure 2 plants-14-01843-f002:**
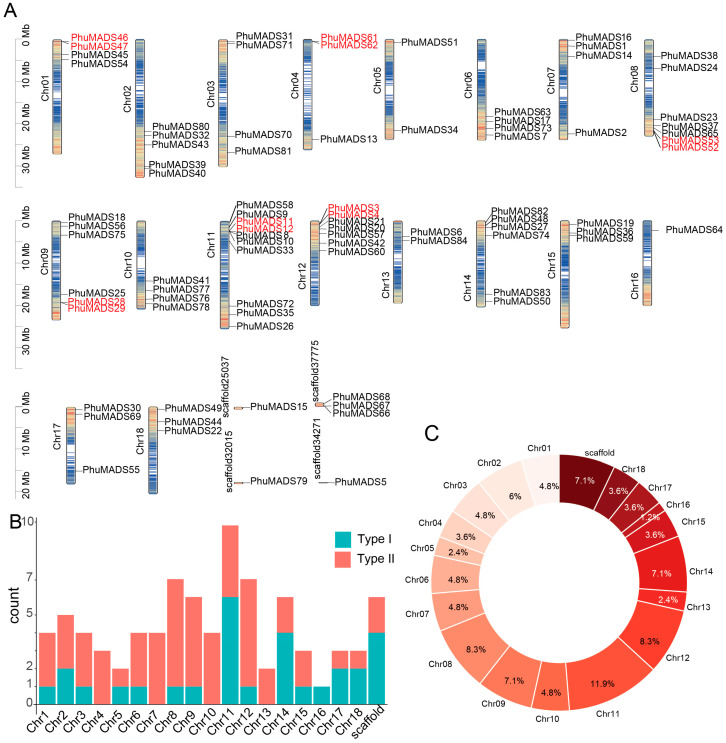
Chromosomal distribution of the *PhuMADS* gene family members. (**A**) Chromosome location of *PhuMADS* genes. (**B**) The number of Type I and Type II *PhuMADS* genes on each chromosome. (**C**) The proportion of *PhuMADS* genes in each chromosome.

**Figure 3 plants-14-01843-f003:**
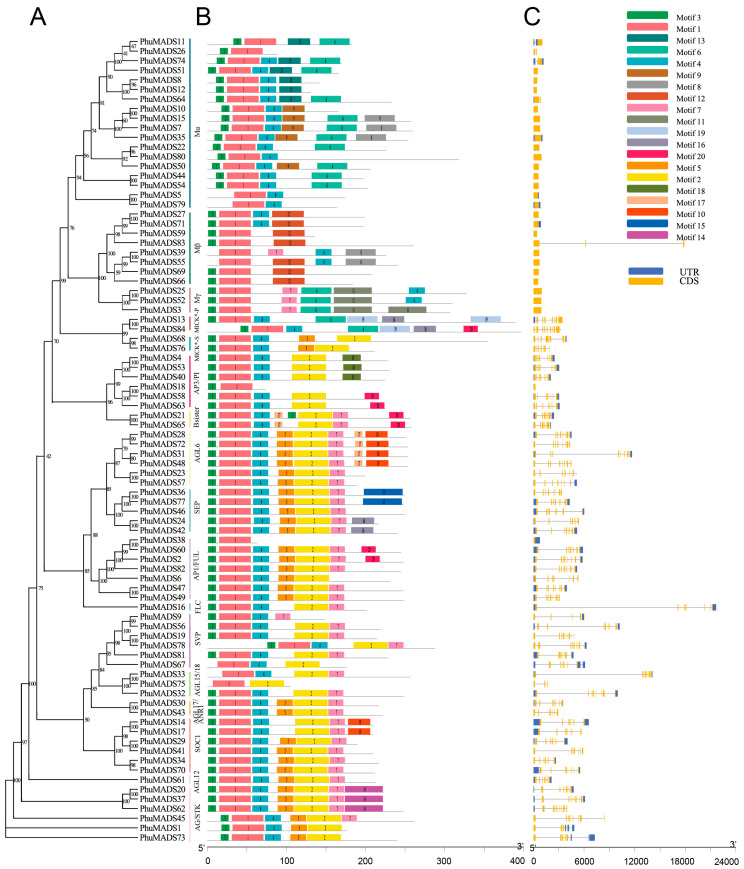
Protein conserved motif and gene structure analysis of *PhuMADS* genes. (**A**) Phylogenetic tree of *PhuMADS*. (**B**) Conserved motif of *PhuMADS* proteins. (**C**) Gene structure of *PhuMADS* genes.

**Figure 4 plants-14-01843-f004:**
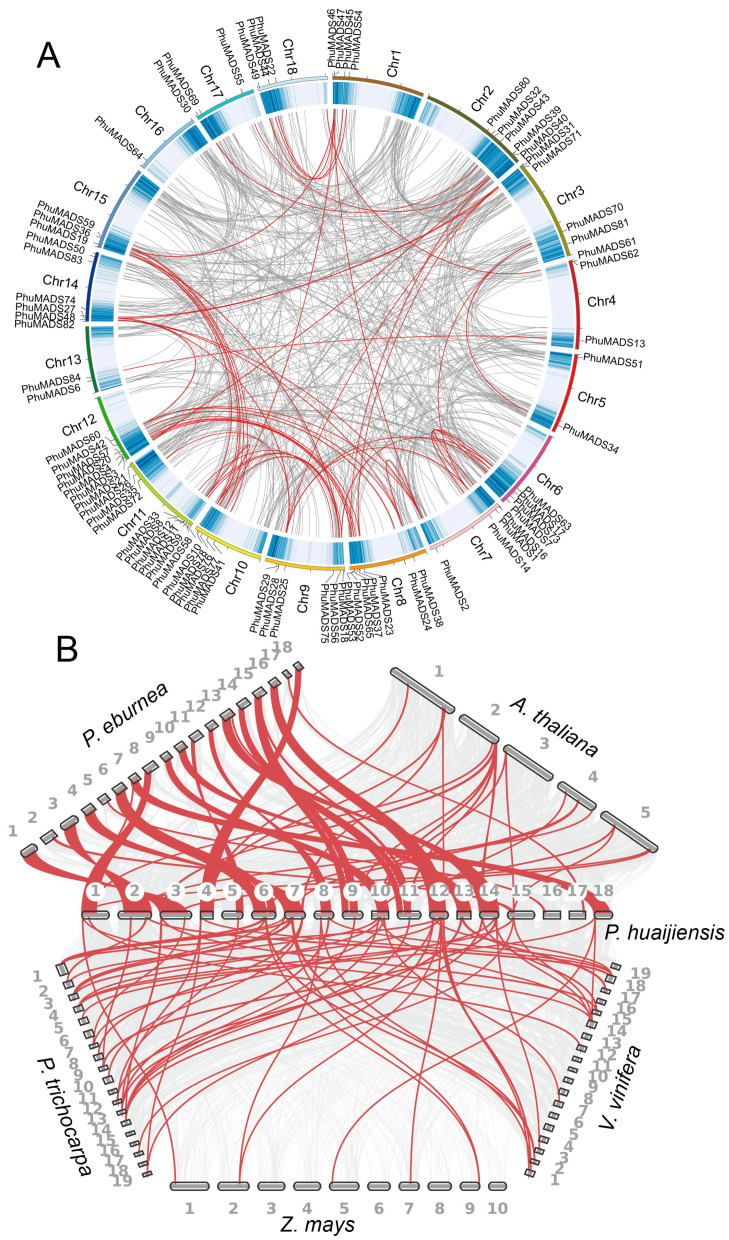
Collinearity analysis of *MADS-box* genes. (**A**) Collinearity analysis of *MADS-box* genes in *P. huaijiensis*. (**B**) Collinearity analysis of *MADS-box* genes between *P. huaijiensis* and *Arabidopsis thaliana*, *Primulina eburnea*, *Populus trichocarpa*, *Vitis vinifera*, *Zea mays*. The gray lines in the background indicate the collinear blocks of all genes, while the red lines highlight the *MADS-box* genes.

**Figure 5 plants-14-01843-f005:**
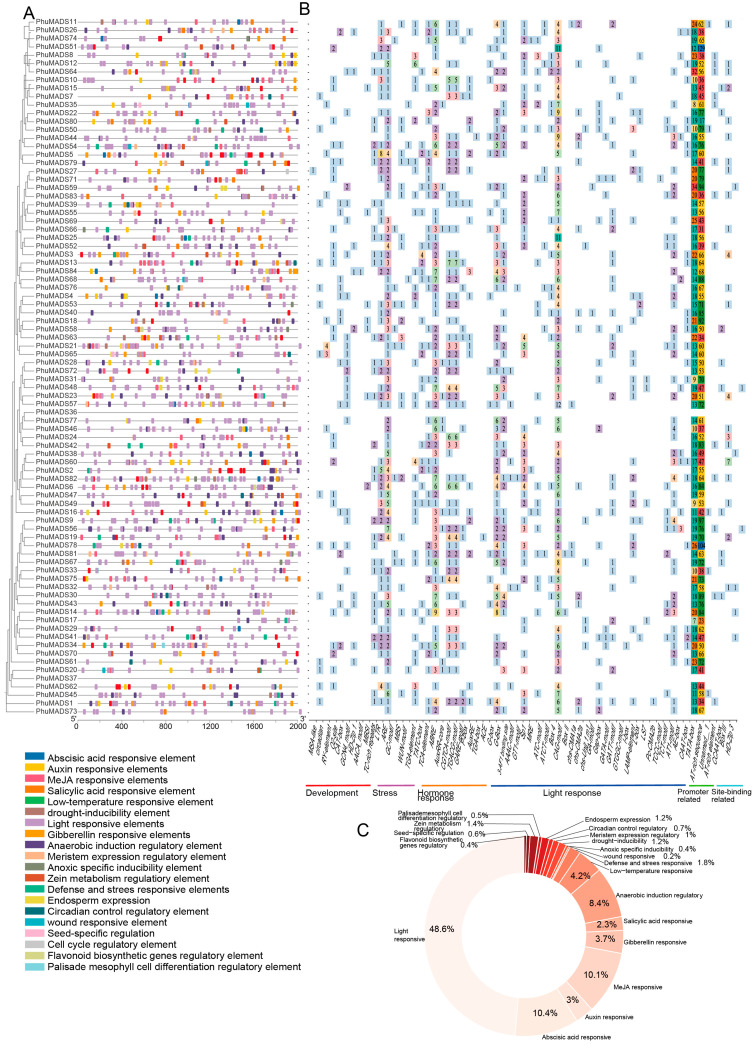
Predicted *cis*-elements in the promoter regions of the *PhuMADS*. (**A**) The distribution of *cis*-acting elements in the *PhuMADS* promoter region (−2000 bp), legend shows the corresponding elements. (**B**) Analysis of *cis*-acting elements of *PhuMADS* gene, numbers in the grid represent the count of element. (**C**) The proportion of the predicated *cis*-regulatory elements among the promoter of *PhuMADS*.

**Figure 6 plants-14-01843-f006:**
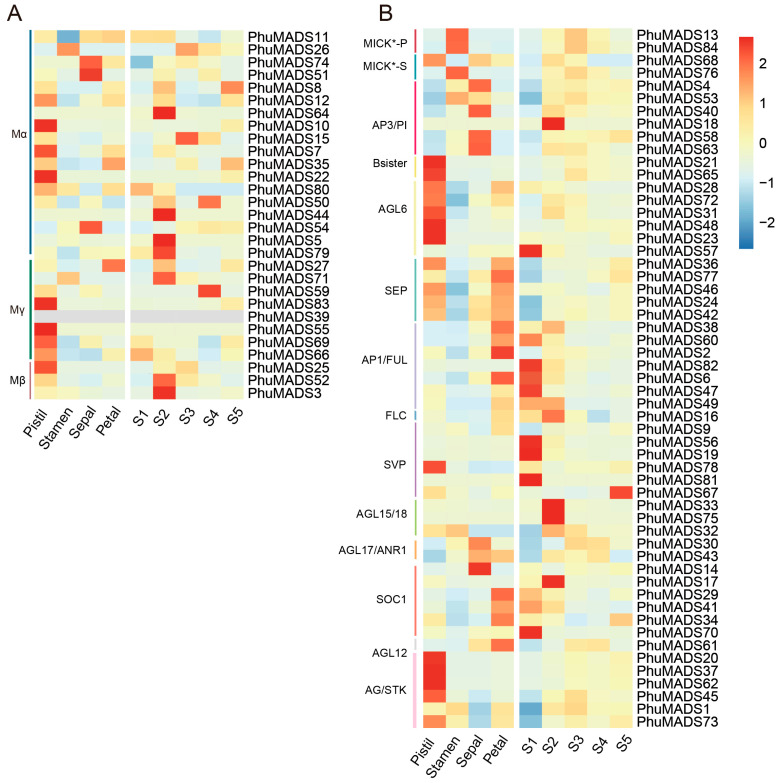
Expression profiles of *MADS-box* genes in different tissues of flowers and developmental stages. (**A**,**B**) indicate the expression profile of type I and type II *PhuMADS* genes. The ratio of the expression levels is presented as log2 fold change (log2 FC) of the FPKM. The letters on the vertical axis indicate the group, S1 to S5 on the horizontal axis indicate the different developmental stages of flower during flowering.

**Figure 7 plants-14-01843-f007:**
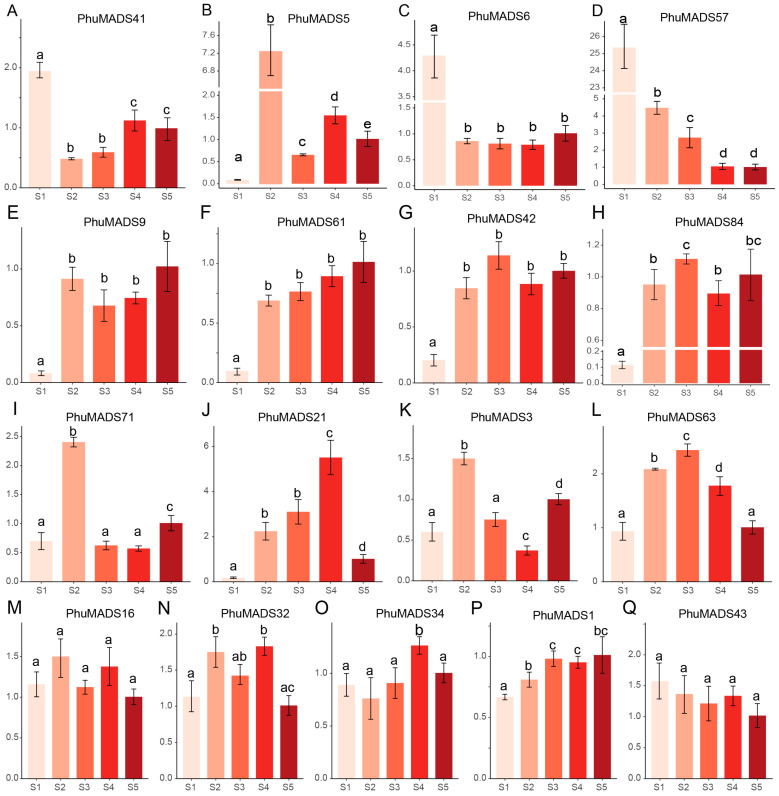
The expression analysis of selected *PhuMADS* genes among different developmental stages. *Y*-axes indicate the relative expression level of each gene. S1 to S5 indicate the different developmental stages of the flower, and the different letters on the bars indicate statistically significant differences at the *p* < 0.05 level.

**Figure 8 plants-14-01843-f008:**
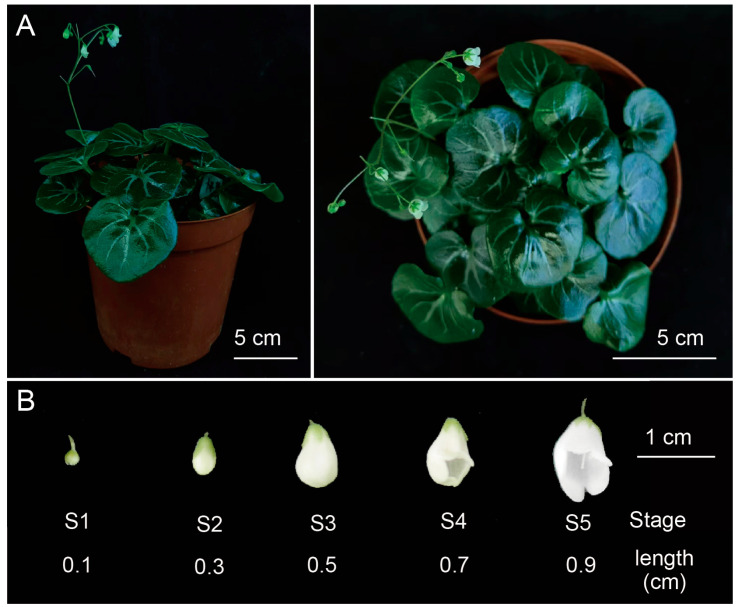
Morphology of *Primulina huaijiensis*. (**A**) The front and the top of the *P. huaijiensis*. (**B**) Samples of flowers across five developmental stages.

## Data Availability

The original contributions presented in this study are included in the article/Appendix A. Further inquiries can be directed to the corresponding author.

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
