# Peer review of "Genome-Wide Identification and Expression Analyses of the MADS-Box Gene During Flowering in Primulina huaijiensis"

_plants, 2025, doi:10.3390/plants14121843_

Round 1
Reviewer 1 Report
Comments and Suggestions for Authors
The manuscript reporting the bioinformatic analysis of MADS gene in Primulina huaijiensis. The manuscript is well structured but need little modification and additional expeimental data.
few comments are:
- From title, please remove Gesneriaceae and use the common name if required; others no need.
- In abstract, a few sentences are too long like line 20-24,
- references cited in intro are too old, please add references from the last five years 2020-2025 with cutting-edge approaches.
- in intro add not famous studies reporting key biological process like development, stress responses
- in M&M statistical analysis missing please add for qRT-PCR
- Please provide clear figures and some fig have inconsistent font like in fig 5b
- in figure 5,6,7,8 please fully explain the fig in legends like what are S1 S2 ..... so on
- what criteria adopted to choose genes for qRT-PCR please mentioned it
- At least 1 gene should be characterised or add transient expression.
Reviewer 2 Report
Comments and Suggestions for Authors
Dear authors:
The findings were derived and obtained using appropriate analytical methods. The obtained results are convincing and clearly presented. I have some suggestions:
- The article’s title should be slightly refined:
“Genome-Wide Identification and Phylogenetic and Expression Analyses of the MADS-box Gene Family in Primulina huaijiensis (Gesneriaceae)”
→ “Genome-Wide Identification and Expression Analyses of the MADS-box Gene Family in Primulina huaijiensis”
- Noted that there is no content included in Table S1. Please add the primer sequences, and don’t forget to include one more column for the GenBank/Accession number of these genes in Table S1
- Noted that Tables S6 and S8 have no title. Please provide.
- Change the citation style of reference: change [authors] to number [1]. References should be indicated by a numeral or numerals in square brackets and numbered in order of appearance —e.g., [1] or [2,3], or [4–6].
For example: Line 36 (Qiu et al., 2022; Lee et al., 2013; Alvarez‐Buylla et al., 2008) → [1–3]
- The number of type Ⅰ and type Ⅱ PhuMADS genes on each chromosome in Fig. 3B should be described and discussed since the author had just mentioned only the total number of them (both typeⅠ and type Ⅱ).
- The sentence should be revised: “To further analyze the orthologous relationships between PhuMADS genes and other representative species …” (Line 268-269) → “To further analyze the orthologous relationships of MADS genes between P. huaijiensis and other representative species …”
7 Reduce the repetition of cited figure data. For example, Fig.7 should be cited only once, at Line 307, which is ok. The subsequent citations (Lines 308 and 312) are unnecessary as the paragraph is discussing the results of the same subject (Fig.7). Additionally, there are multiple (2) panels in Fig.7 labeled as A and B. Please description of what is contained in these panels.
- Authors should include the data of expression profiles of MADS-box genes in different tissues of flowers in Fig.8 (qRT-PCR) as they were presented in Fig.7B (FPKM).
- The section Discussion should be expanded. Authors should discuss the results and findings on how they can be interpreted from the per-spective of previous studies. Their implications should be discussed and future research directions may also be highlighted.
- Please pay special attention to the genes, transcription factors (TFs), and proteins mentioned in the study, using italics to differentiate genes from TFs and proteins. For gene, it pesent in italic. However, protein should not be presented in italics. For example, “MADS-box” (Lines 94, 111…) refers to a protein, therefore, it should not be presented in italics. Please double-check the whole manuscript and revise it.
- It is needed to include one sub-section for Data Analysis (in the section: Materials and Methods), describe what statistical analysis was used and which program was performed MS Excel to analyze/compare the mean differences in data, since the work included the Fig.8 data.
- Represent the manuscripts following the journal format:
- “Capitalize Each Word” in the subsection titles.
For example, “2.8. Expression profiling of PhuMADS genes” → “2.8. Expression Profiling of PhuMADS Genes”
- Section “Reference” should be presented according to the journal’s style and include an abbreviated journal name, year, volume, page range as journal format: “Author 1, A.B.; Author 2, C.D. Title of the article. Abbreviated Journal Name Year, Volume, page range.”
For example, Ref :
“Alvarez-Buylla E R, Pelaz S, Liljegren S J, et al. 2000. An ancestral MADS-box gene duplication occurred before the divergence of plants and animals. Proc Natl Acad Sci U S A 97: 5328-5333.”
→ “Alvarez-Buylla E R, Pelaz S, Liljegren S J, et al. An ancestral MADS-box gene duplication occurred before the divergence of plants and animals. PNAS, 2000, 97, 5328–5333.”
- Reduce font size of 1st affiliation: “1 Jiangxi Provincial Key Laboratory of ex situ Plant Conservation and Utilization, Lushan Botanical Garden, Chinese Academy of Sciences, Jiujiang 332900, China”
- Please consider rearranging the structure of the section manuscript as order: 1. Introduction → 2. Results → 3. Discussion → 4. Materials and Methods → 5. Conclusions
I have marked some above remarks on the manuscript. Please use it for easy tracking and revision.
Best regards,

Reviewer 3 Report
Comments and Suggestions for Authors
- The source of the plant material should be included in the material and method section. It will be great, if authors provide further information about the chosen material and what influence their decision to select this material for the experiment.
- The experimental treatment is not clear. How was the planting done? What were the experimental treatment groups and how was this achieved? Was randomization done? At what stage (in days or period) was the plants sampled?
- I suggest the title be modified to categorically state that the expression analysis was done under flowering conditions, consideration that MADS-box genes have diverse functions.
- Authors should state the significance of the findings in plant breeding and improvement.
- Limitations of the study can be recommended for future studies, aimed as elucidating the function of MDAs-box gene under different environmental conditions.
Round 2
Reviewer 1 Report
Comments and Suggestions for Authors
accept in present form
Author Response
Thank you very much for your efforts in review this manuscript.
Reviewer 2 Report
Comments and Suggestions for Authors
Dear authors:
The authors have made efforts to improve the manuscript. I have some remarks for the revised version:
12. Represent the manuscripts following the journal format:
New comments: need to be revised further.
13. Please consider rearranging the structure of the section manuscript as follows: 1. Introduction → 2. Results → 3. Discussion → 4. Materials and Methods → 5. Conclusions
New comments: The authors have changed by rearranging the sections, as shown in the revised version. However, the numbering of sections was out of order. For instance, (Line 139) “3. Results” → “2. Results” and (Line 328) “4. Discussion” → “3. Discussion”. Moreover, noted that the section “Materials and Methods” (Line 78-139) has not been removed. Please delete it since it was rearranged and now has been presented as section “4. Materials and Methods” (Line 357-456).
Best regards,

Author Response
12. Represent the manuscripts following the journal format:
New comments: need to be revised further.
Response: Thank you very much. We have checked the format and revised the subsection titles, and changed the reference style into the numbered style in last revision response. Please feel free to contact me if any questions occurs.
13. Please consider rearranging the structure of the section manuscript as follows: 1. Introduction → 2. Results → 3. Discussion → 4. Materials and Methods → 5. Conclusions
New comments: The authors have changed by rearranging the sections, as shown in the revised version. However, the numbering of sections was out of order. For instance, (Line 139) “3. Results” → “2. Results” and (Line 328) “4. Discussion” → “3. Discussion”. Moreover, noted that the section “Materials and Methods” (Line 78-139) has not been removed. Please delete it since it was rearranged and now has been presented as section “4. Materials and Methods” (Line 357-456).
Response: Thank you very much. We have revised the number of subsection. Please see the latest revised manuscript.